# Genetic Variants at the *APOE* Locus Predict Cardiometabolic Traits and Metabolic Syndrome: A Taiwan Biobank Study

**DOI:** 10.3390/genes13081366

**Published:** 2022-07-29

**Authors:** Kuan-Hung Yeh, Hsiang-Lin Wan, Ming-Sheng Teng, Hsin-Hua Chou, Lung-An Hsu, Yu-Lin Ko

**Affiliations:** 1Cardiovascular Center and Division of Cardiology, Department of Internal Medicine, Taipei Tzu Chi Hospital, Buddhist Tzu Chi Medical Foundation, New Taipei City 23142, Taiwan; ufddsykh@ms15.hinet.net (K.-H.Y.); chouhhtw@gmail.com (H.-H.C.); 2School of Medicine, Tzu Chi University, Hualien 97004, Taiwan; 3Division of Hematology/Oncology, Department of Internal Medicine, Taipei Tzu Chi Hospital, Buddhist Tzu Chi Medical Foundation, New Taipei City 23142, Taiwan; wanhl@tzuchi.com.tw; 4Department of Research, Taipei Tzu Chi Hospital, Buddhist Tzu Chi Medical Foundation, New Taipei City 23142, Taiwan; vincent@tzuchi.com.tw; 5The First Cardiovascular Division, Department of Internal Medicine, Chang Gung Memorial Hospital and Chang Gung University College of Medicine, Taoyuan 33305, Taiwan; hsula@cgmh.org.tw

**Keywords:** *APOE* locus, *CLPTM1*, *APOC1*, lipid profile, metabolic syndrome, serum albumin level

## Abstract

Several apolipoprotein genes are located at the *APOE* locus on chromosome 19q13.32. This study explored the genetic determinants of cardiometabolic traits and metabolic syndrome at the *APOE* locus in a Taiwanese population. A total of 81,387 Taiwan Biobank (TWB) participants were enrolled to undergo genotype–phenotype analysis using data from the Axiom Genome-Wide CHB arrays. Regional association analysis with conditional analysis revealed lead single-nucleotide variations (SNVs) at the *APOE* locus: *APOE* rs7412 and rs429358 for total, low-density lipoprotein (LDL), and high-density lipoprotein (HDL) cholesterol levels; *CLPTM1* rs3786505 and rs11672748 for LDL and HDL cholesterol levels; and *APOC1* rs438811 and *APOE-APOC1* rs439401 for serum triglyceride levels. Genotype–phenotype association analysis revealed a significant association of rs429358 and rs438811 with metabolic syndrome and of rs7412, rs438811, and rs439401 with serum albumin levels (*p* < 0.0015). Stepwise regression analysis indicated that *CLPTM1* variants were independently associated with LDL and HDL cholesterol levels (*p* = 3.10 × 10^−15^ for rs3786505 and *p* = 1.48 × 10^−15^ for rs11672748, respectively). *APOE* rs429358 and *APOC1* rs438811 were also independently associated with metabolic syndrome (*p* = 2.29 × 10^−14^) and serum albumin levels (*p* = 3.80 × 10^−6^), respectively. In conclusion, in addition to *APOE* variants, *CLPTM1* is a novel candidate locus for LDL and HDL cholesterol levels at the *APOE* gene region in Taiwan. Our data also indicated that *APOE* and *APOC1* variants were independently associated with metabolic syndrome and serum albumin levels, respectively. These results revealed the crucial role of genetic variants at the *APOE* locus in predicting cardiometabolic traits and metabolic syndrome.

## 1. Introduction

The *APOE* gene is located within the *APOE-C1-C4-C2* gene cluster on chromosome 19q13.32, which encodes four amphipathic apolipoproteins and two hepatic control regions that regulate the hepatic expression of these genes [1,2]. Several other genes, such as *TOMM40* and *CLPTM1*, that are not directly associated with lipid metabolism are also located close to *APOE*. This genomic locus is characterized by strong linkage disequilibrium (LD) between different polymorphic genotypes in this region, which tend to be coinherited faithfully. Thus, genome-wide association studies (GWASs) have indicated that most of the gene loci on the *APOE* locus region are associated with multiple lipid variables. By the NHGRI-EBI GWAS Catalog [3], a publicly available resource for GWASs, the major contribution of the *APOE* locus variants includes not only the regulation of low-density lipoprotein (LDL)-related traits but also their significant associations with triglyceride and high-density lipoprotein (HDL) cholesterol levels. 

Human apolipoprotein E (APOE), synthesized and secreted by hepatocytes, acts by binding to its own receptor and the LDL receptor. It is the main ligand for the clearance of very-low-density lipoproteins (VLDLs) and chylomicron remnants and affects the circulating concentration of lipoproteins and plasma levels of cholesterol and triglycerides [4]. APOE participates in the reverse cholesterol transport mechanism and mediates the hepatic uptake of triglyceride-rich lipoprotein (TRL). *APOE* variants have been reported in different types of dyslipidemias, such as autosomal-dominant hypercholesterolemia, familial combined hyperlipidemia, familial dysbetalipoproteinemia, and lipoprotein glomerulopathy [5]. Dyslipidemia is a modifiable risk factor of atherosclerosis, which is currently the most important pathological mechanism leading to the development of cardiovascular disease [6,7,8]. The most important forms of atherosclerotic cardiovascular disease are coronary heart disease, cerebrovascular disease, and peripheral arterial disease. Three APOE isoforms, created by the ε2, ε3, and ε4 alleles with two single-base changes (*APOE* rs7412 and *APOE* rs429358) in the coding region of *APOE*, are the most well-defined common variants that determine plasma lipid levels, coronary risk, and Alzheimer disease [9,10,11,12].

Three other apolipoprotein genes are located within the *APOE-C1-C4-C2* gene cluster. 

Apolipoprotein C1 is a constituent of TRL and HDL and acts on the exchanges between lipoprotein classes, leading to decreased LDL cholesterol levels [13,14,15]. ApoC1 also modulates the activities of several enzymes, such as the activation of lecithin cholesterol acyl transferase and inhibition of cholesterol ester transfer protein and lipoprotein lipase (LPL), resulting in the control of serum triglyceride and HDL cholesterol levels [13,16,17]. Apolipoprotein C-IV (*APOC4*), a highly conserved lipid-binding protein associated mainly with VLDL particles, plays a vital role in triglyceride metabolism [18]. Transgenic mice with *APOC4* overexpression exhibit elevated triglyceride levels [19,20]. Apolipoprotein C-II (apoC-II) is a small, exchangeable apolipoprotein found on TRL and plays a critical role in TRL metabolism by acting as an essential cofactor of LPL [21]. Both a deficiency and an excess of apoC-II are associated with reduced LPL activity and hypertriglyceridemia [22,23].

Two other genes that are not directly associated with lipid metabolism and located close to *APOE* are also candidate genes for lipid profile. *TOMM40* encodes M40 (translocase of the outer mitochondrial membrane, 40 kD), which forms the channel subunit of a multisubunit complex of the outer mitochondrial membrane pore subunit. A study involving postmortem brain analysis concluded that by recognizing and allowing the importation of nuclear-encoded proteins, M40 can induce dynamic mitochondrial dysfunction in neurons to increase the risk of Alzheimer disease along with the upregulation of *TOMM40* messenger RNA in the frontal lobe of a postmortem Alzheimer disease brain compared with controls [24]. GWASs have demonstrated a significant association between *TOMM40* variants and body mass indices, LDL-C levels, and Alzheimer disease [25,26,27]. *CLPTM1* was originally identified as the causative gene mutation in a patient family with cleft lip and palate [28]. Clptm1 is a γ-aminobutyric acid A receptor (GABA_A_R)-associated protein [29], and GWASs in Europeans have demonstrated that *CLPTM1* variants are associated with multiple lipid traits [30,31]. However, their role in Asian populations remains unknown. We previously reported that *APOE* variants interact with C-reactive protein to regulate triglyceride levels; thus, triglyceride concentration is influenced by both the genetic background of the *APOE* locus and the inflammatory status of a patient [32]. 

The Taiwan Biobank (TWB) is a large-scale, population-based cohort study recruiting volunteers aged between 30 and 70 years with no history of cancer [33]. Because of multiple genes being situated at chromosome 19q13.32, we investigated the genetic determinants of cardiometabolic traits and metabolic syndrome by using regional association analysis in TWB participants.

## 2. Participants and Methods

### 2.1. TWB Participants

The TWB is a population-based research consortium. The study population is comprised of 107,494 TWB Han Chinese participants who had no history of cancer and were recruited between 2008 and 2020 from centers across Taiwan. We excluded 26,107 participants because of (1) fasting for <6 h (2862 participants), (2) no imputation data (12,289 participants), and (3) quality control (QC) for the GWAS with identity by descent PI_HAT > 0.187 (10,956 participants). Figure 1 depicts the flowchart of participant enrollment. We also excluded participants with a history of gout, hypertension, hyperlipidemia, and diabetes mellitus as indicated by their serum uric acid level, blood pressure status, and lipid and glucose metabolism parameters, respectively. Definitions of hypertension, diabetes mellitus, obesity, hyperlipidemia, and current smoking are provided in Supplementary Method 1. This study was approved by the Research Ethics Committee of Taipei Tzu Chi Hospital, Buddhist Tzu Chi Medical Foundation (approval number: 05-X04-007), and the Ethics and Governance Council of the Taiwan Biobank (approval numbers: TWBR10507-02 and TWBR10611-03). Each participant signed an informed consent form before participating in the study.

### 2.2. Genomic DNA Extraction and Genotyping

By using a PerkinElmer Chemagic 360 instrument (PerkinElmer, Waltham, MA, USA), genomic DNA was extracted after blood sampling. Single-nucleotide variation (SNV; formerly referred to as “single-nucleotide polymorphism”) genotyping was performed using custom TWB chips on the Axiom Genome-Wide Array Plate System (Affymetrix, Santa Clara, CA, USA). 

### 2.3. Clinical Phenotypes and Laboratory Examinations

Clinical phenotypes used for the study were body mass index (BMI), waist–hip ratio, waist circumference, and systolic, diastolic, and mean blood pressures. We also collected the following biochemical data: fasting plasma glucose and glycated hemoglobin (HbA1c) levels for glucose metabolism; total, LDL, and HDL cholesterol and triglyceride levels for lipid profile; and aspartate aminotransferase (AST), alanine aminotransferase (ALT), γ-glutamyl transferase (γ-GT), albumin, total bilirubin, serum creatinine, and uric acid levels for liver and renal functional tests. BMI and estimated glomerular filtration rate (eGFR) were calculated as reported previously [34]. White and red blood cell counts, platelet counts, hematocrit, and hemoglobin levels were used for the hematological analysis. Because urine creatinine levels were unavailable, only the spot urine albumin level was used to evaluate albuminuria.

### 2.4. Regional Association Analysis with Conditional Analysis 

To determine which variant was the lead SNV around the *APOE* gene region for lipid profile, we conducted a regional association analysis by including TWB participants enrolled after the exclusion criteria were applied, as reported previously [34,35]. In brief, QC was performed for GWAS using the Axiom Genome-Wide CHB 1 and CHB 2 Array Plates (Affymetrix), each comprising 611,656 and 640,160 SNVs, respectively. Genome-wide genotype imputation was performed using SHAPEIT and IMPUTE2, and the East Asian population from the 1000 Genome Project Phase 3 study was used as the reference panel. With imputation, QC was then performed by filtering SNVs with an IMPUTE2 imputation quality score of >0.3. All samples enrolled for the analysis had a call rate of ≥97%. The indels were removed using VCFtools. For SNV QC, an SNV call rate of <97%, a minor allele frequency of <0.01, and the violation of the Hardy–Weinberg equilibrium (*p* < 10^−6^) were the criteria for exclusion from subsequent analyses. After QC, in the *APOE* region, 325 SNVs at positions between 45.2 and 45.6 Mb on chromosome 19q13.32 were enrolled for illustration and analysis.

### 2.5. Statistical Analysis

By using the Kolmogorov–Smirnov test, continuous variables were tested for normal distribution. All variables were not compatible with normal distribution and were expressed as medians and interquartile ranges. Categorical data were presented as percentages. Lipid profile and urine albumin levels were logarithmically transformed to achieve adherence to a normality assumption before being examined using analysis of variance and regression. A general linear regression was used to analyze the association between the studied phenotypes and investigated genotypes after adjustment for age, sex, BMI, and current smoking status. To evaluate the independent effect of the investigated genotypes on the risk of lifestyle and atherosclerotic risk factors, we used multiple logistic regression analysis. Stepwise multivariable linear regression was then performed to determine the independent correlates of lipid profile and serum albumin levels. Regional association analysis was performed using the PLINK software package. Genome-wide significance was defined as *p* < 5 × 10^−8^. In genotype–phenotype association analysis, the Bonferroni method was used to correct for multiple comparisons where applicable on the basis of the 33 traits analyzed (0.05/33 = 0.0015). LDmatrix software (https://analysistools.nci.nih.gov/LDlink/?tab=ldmatrix, accessed on 12 January 2021) was used to analyze LD. All statistical analyses were performed using SPSS (version 22; SPSS, Chicago, IL, USA).

## 3. Results

### 3.1. Baseline Characteristics

The TWB participants’ clinical, demographic, and laboratory data stratified by sex are presented in Table 1. Compared with female participants, male participants had significantly higher BMI, waist circumference, and waist-to-hip ratio; systolic, diastolic, and mean blood pressures; circulating levels of LDL cholesterol, triglyceride, fasting plasma glucose; uric acid and creatinine; liver injury and function parameters of AST, ALT, γ-GT, albumin, and total bilirubin levels; and hematocrit, red blood cell count, and hemoglobin (all *p <* 0.0001). By contrast, total cholesterol and HDL cholesterol levels, eGFR, albuminuria, and platelet counts (all *p* < 0.0001) were higher in women than in men.

### 3.2. Regional Association Plots

Regional association plots were made to determine the association of genetic variants around the *APOE* gene region at positions 45.2–45.6 Mb on chromosome 19q13.32 for the lipid profile (Figure 2). The lead SNVs with genome-wide significant associations included *APOE* rs7412 for total, LDL, and HDL cholesterol levels and *APOC1* rs438811 for serum triglyceride levels (*p* < 10^−307^, *p* < 10^−307^, *p* = 2.37 × 10^−44^ and *p* = 1.61 × 10^−85^, respectively). We performed sequential conditional analysis for each lipid trait, and the lead SNVs with genome-wide significant associations were *APOE* rs429358 for total cholesterol levels (*p* = 2.38 × 10^−71^), *APOE* rs429358 and *CLPTM1* rs3786505 for LDL cholesterol levels (*p* = 5.81 × 10^−120^ and *p* = 1.67 × 10^−14^, respectively), *APOE* rs429358 and *CLPTM1* rs11672748 for HDL cholesterol levels (*p* = 1.21 × 10^−43^ and *p* = 6.89 × 10^−16^, respectively), and *APOE-APOC1* rs439401 for serum triglyceride levels (*p* = 2.59 × 10^−66^; Table 2). Nearly complete LD was noted between rs11672748 and rs3786505 (r^2^ > 0.99), and both variants had weak LD with other lead SNVs (all r^2^ < 0.015). Weak LD was also noted between rs429358, rs7412, and rs439401 (all r^2^ < 0.2), and these three SNVs were in moderate LD with rs438811 (r^2^ between 0.310 and 0.432). The results are presented in Figure 3.

### 3.3. Genotype–Phenotype Association Analysis for the APOE Locus Lead SNVs 

We tested the association of *APOE* locus lead SNVs with clinical, metabolic, and biochemical phenotypes; hematological parameters; lifestyle habits; and risk factors for atherosclerosis (Appendix A). For the lipid profile, after adjustment for age, sex, current smoking status, and BMI, genome-wide significant associations were noted between rs7412, rs429358, and rs439401 and all four lipid traits, between rs438811 and total and LDL cholesterol and triglyceride levels, and between rs11672748 and rs3786505 and HDL and LDL cholesterol levels. Genome-wide significant associations were also noted between rs429358 and rs438811 and the risk of metabolic syndrome. With Bonferroni correction, we also observed a significant association (*p* < 0.0015) between rs7412, rs438811, and rs439401 and serum albumin levels, between rs429358 and eGFR and ALT levels, between rs438811 and waist–hip ratio and ALT levels, and between rs11672748 and rs3786505 and erythrocyte counts.

### 3.4. Stepwise Linear Regression Analysis between the APOE Locus-Lead SNVs and Lipid Profile 

Stepwise linear regression analyses using age, sex, BMI, current smoking status, and *APOE* locus lead SNVs revealed that these variants together contributed to 2.39%, 8.79%, 0.41%, and 0.47% of the total variation in serum total, LDL, and HDL cholesterol and triglyceride levels, respectively (Table 3). Moreover, the *APOC1* rs438811 variant was independently associated with serum albumin levels (Appendix A). Stepwise logistic regression analysis demonstrated that *APOE* rs429358 is independently associated with metabolic syndrome (odds ratio: 1.20 for each C-allele, 95% confidence interval: 1.15–1.26, *p* = 2.29 × 10^−14^; Table 4).

## 4. Discussion

We investigated the associations between genetic variants around the *APOE* locus, cardiometabolic traits, and metabolic syndrome in a Taiwanese population. We demonstrated that *APOE* ɛ2, ɛ3, and ɛ4 variants, defined by the *APOE* rs7412 and *APOE* rs429358 genotypes, were the strongest genetic determinants in the *APOE* locus of total, LDL, and HDL cholesterol levels in Taiwan. Both *APOE* and *CLPTM1* variants were independently associated with LDL and HDL cholesterol levels, and the genetic variants between the *APOE* and *APOC1* gene region, including rs429358 and rs438811, were lead SNVs for se-rum triglyceride levels. Our data also provide evidence for the association of *APOE* locus variants with metabolic syndrome and serum albumin levels. To the best of our knowledge, this is the first report revealing *CLPTM1* as a novel candidate locus for LDL and HDL cholesterol levels independent of the *APOE* variants in an Asian population. Both *CLPTM1* variants had weak LD with the lead SNVs in the *APOE* and *APOC1* genes. *APOE* is a fascinating multifunctional apolipoprotein that affects cardiovascular and neurological health through common variants [4,36]. Our results suggest that a more complete genotyping in this chromosomal region may help to elucidate more of the crucial roles of the *APOE* locus variants on cardiometabolic and neurological disorders.

### 4.1. APOE Variants and Lipid Profile

APOE is a key regulator of plasma lipid levels, and the Apoe-/- mouse model is the most widely used animal model of atherosclerosis, with markedly elevated plasma cholesterol levels [4,37]. The ApoE3 isoform possesses both the lipid-binding ability and affinity for the LDL receptor to mediate appropriate lipolytic processing and endocytosis of TRL remnant particles and exists as the parent form of three common APOE isoforms with normal plasma lipid levels [11]. ApoE2 and apoE4 differ from apoE3 by the single amino acid substitution with Arg158Cys (rs429358) for apoE2 which is located near the LDL receptor recognition site and Cys112Arg (rs7412) for apoE4, which affected the organization and stability of both the N-terminal and C-terminal domains of APOE. ApoE2 exhibits impaired binding to the LDL receptor and an inability to promote clearance of TRL remnant particles. ApoE4 results in enhancing the binding ability of VLDL particles and impairs lipolytic processing in the circulation, which is associated with a higher VLDL cholesterol/HDL cholesterol ratio as a more proatherogenic lipoprotein cholesterol distribution [11]. In a meta-analysis of more than 800,000 individuals from BioBank Japan, UK Biobank, and FinnGen, a strong association between rs7412 and LDL-C levels was noted, with a very low *p* value of 3 × 10^−3040^ [38]. Our data also reveal that the *APOE* ɛ2, ɛ3, and ɛ4 variants defined by the *APOE* rs7412 and *APOE* rs429358 genotypes were the strongest genetic determinants of LDL cholesterol levels in our Taiwanese cohort, which together contributed to 8.72% of LDL cholesterol levels.

### 4.2. CLPTM1 Polymorphism and Serum Lipid Levels

Clptm1 is a multi-pass transmembrane protein whose biological function has not been fully elucidated. *CLPTM1* variants have been suggested to be one of the genetic risk factors of non-syndromic oral clefts [39]. Genetic variants around the *CLPTM1* have been associated with episodic memory performance [40]. The associations between *CLPTM1* gene variants and expression and Alzheimer disease have been observed in genome-wide and transcriptome-wide association studies [27,40,41]. Several GWASs have revealed *CLPTM1* variants associated with various lipid traits and apolipoprotein B levels in European populations [30,31,42,43]. Our data also indicate that *CLPTM1* variants were associated with LDL and HDL cholesterol levels independent of *APOE* variants in a Taiwanese cohort. These results support the crucial role of *CLPTM1* in lipid metabolism. Clptm1 has been demonstrated to be a pan-GABAA receptor-associated protein that interacts with multiple subunits and traps GABAA receptors in the endoplasmic reticulum and Golgi apparatus to scale phasic and tonic inhibitory transmission and modulate activity-induced inhibitory homeostasis [29,44,45]. The γ-aminobutyric acid (GABA) is the principal inhibitory neurotransmitter that acts on GABA receptors and modulates cholesterol metabolism in macrophages to increase cholesterol efflux and inhibit the formation of human macrophage-derived foam cells [46]. Through integrative analysis of multiomics data from the Avon Longitudinal Study of Parents and Children and replicated in the TwinsUK study, *CLPTM1* is identified to be associated with LDL cholesterol levels by the proposed powerful adaptive gene-based test integrating information from gene expression, methylation, and enhancer–promoter interactions with the signal driven by both low-frequency and common variants [47]. Thus, further study is needed to unravel the molecular mechanism(s) for the associations between *CLPTM1* variants and lipid profiles.

### 4.3. APOE-APOC1 Polymorphisms and Serum Triglyceride Levels 

Both rs438811 and rs439401 variants, located between *APOE* and *APOC1*, have been associated with serum lipid levels in several GWASs [42,48,49,50]. Apolipoprotein C1 (apoC1) is the smallest of all apolipoproteins and is associated with TRLs and HDL and exchanges between lipoprotein classes. It acts on lipoprotein receptors by inhibiting binding mediated by APOE and modulating the activities of several enzymes [51]. The rs438811 variant is in moderate LD with *APOE* rs7412 and rs429358 variants and is associated with total, LDL cholesterol, and triglyceride levels, whereas the rs439401 variant is in weak LD with the *APOE* rs7412 and rs429358 variants and is associated with all four lipid profile variables. For serum triglyceride levels, these two variants were the most crucial genetic determinants in the *APOE* locus in TWB participants. 

### 4.4. Association between APOE Gene Region Variants and Metabolic Syndrome 

Metabolic syndrome involves a complex of interrelated cardiometabolic derangements linked to insulin resistance and chronic inflammation and is characterized by clustering of cardiovascular risk factors, including hypertension, dyslipidemia, central obesity, and glucose intolerance [52]. Metabolic syndrome is also associated with higher risks of type 2 diabetes mellitus, nonalcoholic fatty liver disease, atherosclerotic cardiovascular disease, ischemic stroke, and mortality [48,53,54,55,56]. A high prevalence of metabolic syndrome has been observed in the Asia-Pacific region, including Taiwan [57]. Strong genetic components interacting with environmental factors have been associated with the risk of metabolic syndrome. Elucidating genetic risk factors for metabolic syndrome remains a challenging task in the prevention and management of the syndrome complex. Studies for the association between genetic variants on the *APOE* locus and metabolic syndrome are not consistent with some GWASs; some genetic association studies have demonstrated *APOE* or *APOC1* variants associated with the risk of metabolic syndrome [58,59,60,61], whereas others have reported no evidence of this association [62,63,64]. Our results reveal an independent association between *APOE* rs429358 genotypes and metabolic syndrome. One study suggested an association between *APOE* rs429358 genotypes and metabolic syndrome, which became nonsignificant after Bonferroni correction [64]. Thus, additional studies are required to elucidate the role of *APOE* variants, especially rs429358, in the risk of metabolic syndrome. 

### 4.5. Association between APOE Gene Region Variants and Serum Albumin Levels

Albumin is a multifaceted protein synthesized in hepatocytes involving anti-inflammatory, antioxidant, anticoagulant, and antiplatelet aggregation activities, and serum albumin is the most abundant human plasma protein [65]. In our previous study on TWB participants, we found pleiotropic effects of common and rare *GCKR* exonic mutations on cardiometabolic traits, including serum albumin levels [34]. In a cross-population atlas of genetic associations for 220 human phenotypes, using 179,000 BioBank Japan participants and a meta-analysis of studies involving a total of 628,000 UK Biobank and FinnGen participants, Sakaue et al. [38] also revealed the *APOE/APOC1* region as a novel locus for serum albumin levels with *APOC1* rs438811 as the candidate lead SNV. This is consistent with our finding that the rs438811 genotype was significantly associated with serum albumin levels independent of *GCKR* variants.

## 5. Limitations

This study has several limitations. First, we used genome-wide genotyping arrays with imputation data; however, whole-exome sequencing or even whole-genome sequencing is more powerful in detecting whether other functional variants are more important genetic determinants for cardiometabolic traits and metabolic syndrome in the *APOE* locus. Second, ethnic genetic heterogeneity in the genetic association studies suggests that our data may not be applicable to other ethnic populations. Third, not all genetic associations reached genome-wide significance in our analysis, and studies with larger sample sizes and meta-analyses including transethnic population studies may help validate our results.

## 6. Conclusions

Our data reveal that *APOE* ɛ2, ɛ3, and ɛ4 variants were the strongest genetic determinants of total, LDL, and HDL cholesterol levels in the *APOE* locus and *CLPTM1* is a candidate locus for LDL-C and HDL-C levels, independent of other gene variants around the *APOE* gene region. The association between *APOE* and *APOC1* variants and metabolic syndrome and serum albumin levels further supported the crucial role of *APOE* region variants in cardiometabolic disorders in a Taiwanese population.

## Figures and Tables

**Figure 1 genes-13-01366-f001:**
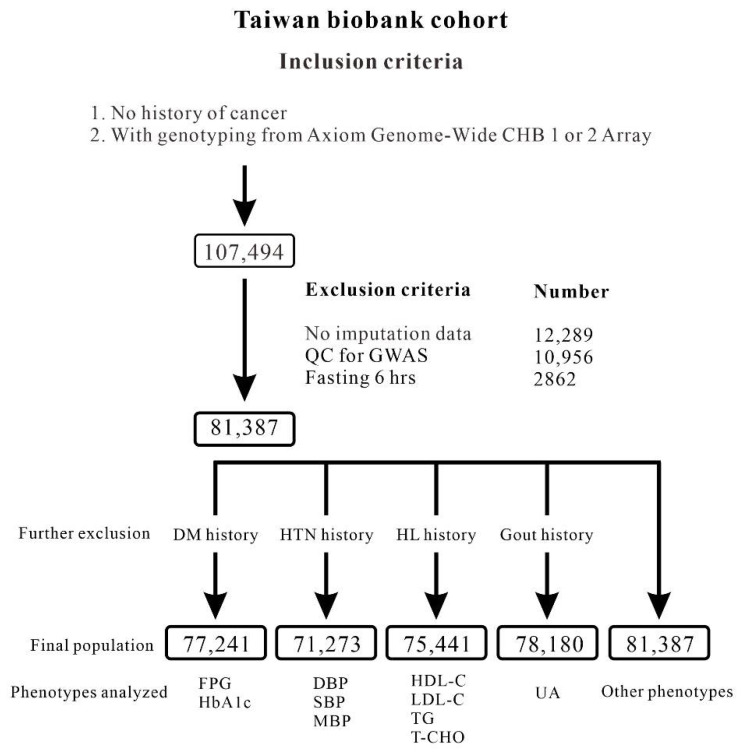
Flowchart of the Taiwan Biobank participants.

**Figure 2 genes-13-01366-f002:**
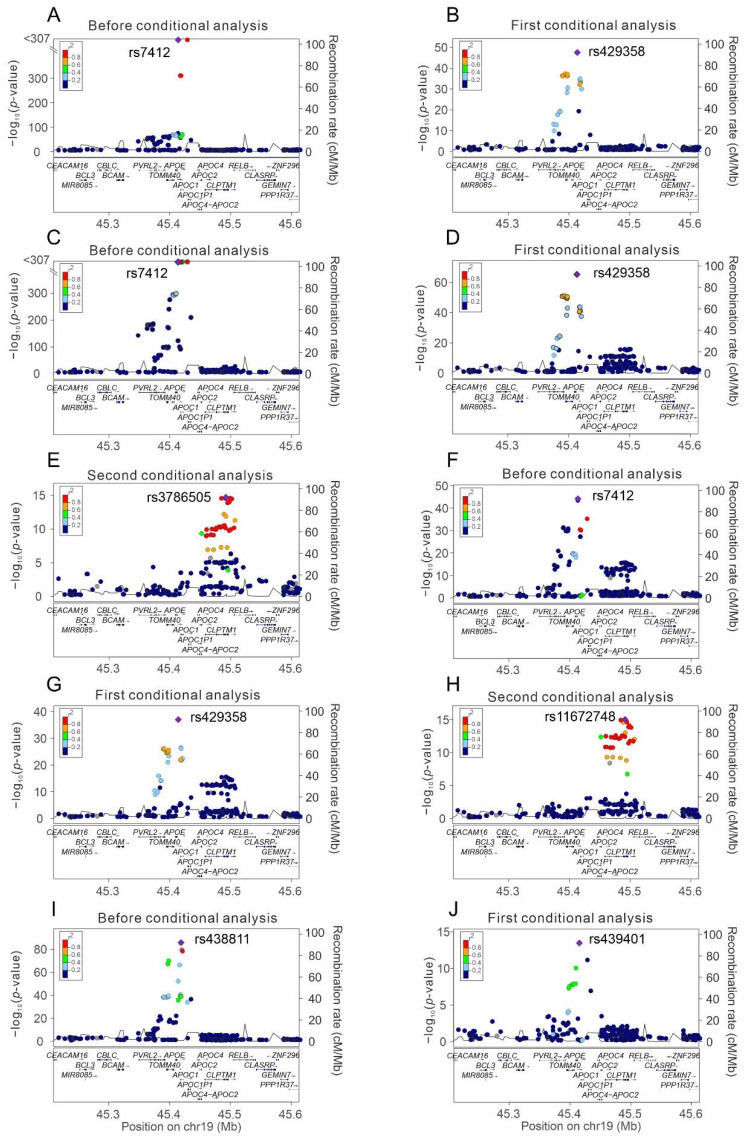
Regional association analysis for genetic variants around the *APOE* gene region and serum lipid levels. (**A**,**B**) For total cholesterol levels, (**C**–**E**) for LDL cholesterol levels, (**F**–**H**) for HDL cholesterol levels, and (**I**,**J**) for triglyceride levels.

**Figure 3 genes-13-01366-f003:**
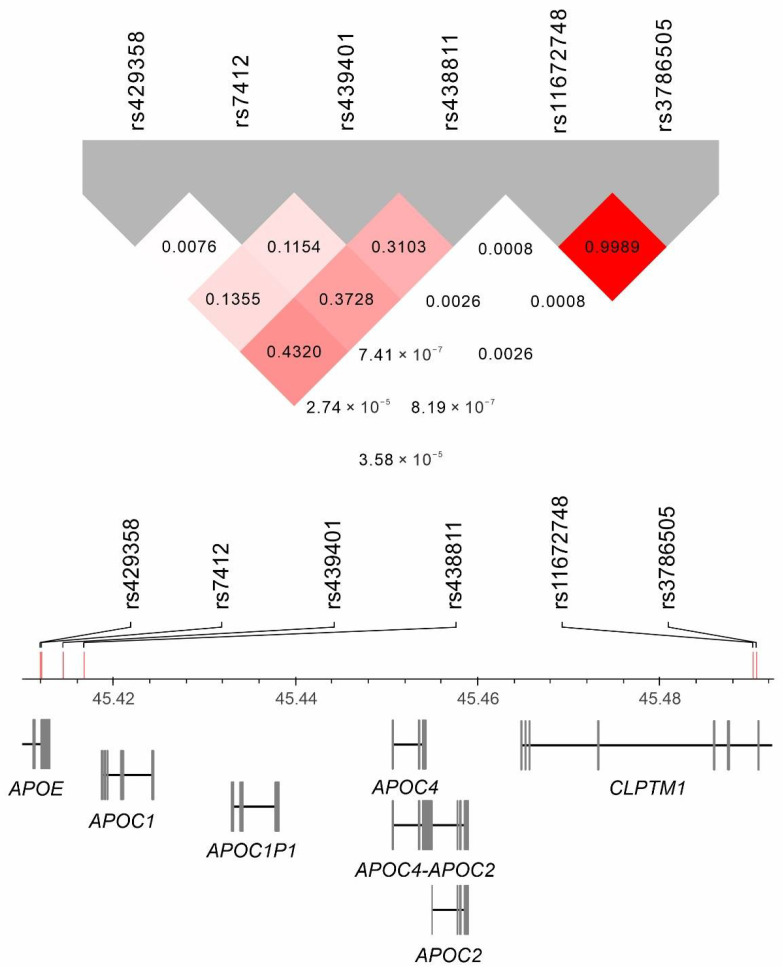
Linkage disequilibrium of *APOE* gene region lead single-nucleotide variants.

**Table 1 genes-13-01366-t001:** Baseline characteristics of study patients by sex.

Clinical and Laboratory Parameters	Total	Male	Female
Number	81,387	29,487	51,900
Anthropology			
Age (years)	51.0 (41.0–59.0)	51.0 (41.0–60.0)	51.0 (41.0–59.0) ***
Waist circumference (cm)	83.0 (76.0–89.5)	87.5 (82.0–93.5)	80.0 (74.0–86.5) ***
Waist–hip ratio	0.87 (0.82–0.91)	0.90 (0.86–0.94)	0.84 (0.80–0.89) ***
Body mass index (kg/m^2^)	23.78 (21.58–26.30)	25.0 (23.0–27.3)	23.0 (21.0–25.5) ***
Blood Pressure			
Systolic BP ^†^ (mmHg)	115.0 (105.0–126.7)	121.0 (111.7–131.5)	111.0 (102.0–123.0) ***
Diastolic BP ^†^ (mmHg)	71.0 (65.0–79.0)	76.5 (70.0–83.0)	69.0 (62.7–76.0) ***
Mean BP ^†^ (mmHg)	86.0 (78.7–94.3)	91.2 (84.3–98.7)	83.2 (76.3–91.3) ***
Lipid profile			
Total cholesterol # (mg/dL)	193.0 (171.0–216.0)	190.0 (168.0–213.0)	194.0 (172.0–218.0) ***
HDL cholesterol # (mg/dL)	53.0 (45.0–63.0)	47.0 (40.0–54.0)	57.0 (49.0–66.0) ***
LDL cholesterol # (mg/dL)	119.0 (99.0–140.0)	121.0 (101.0–142.0)	118.0 (98.0–140.0)
Triglyceride # (mg/dL)	90.0 (63.0–132.0)	107.0 (74.0–156.0)	83.0 (59.0–119.0) ***
Glucose metabolism			
Fasting plasma glucose ^††^ (mg/dL)	92.0 (87.0–97.0)	94.0 (89.0–99.0)	90.0 (86.0–95.0) ***
HbA1c ^††^ (%)	5.6 (5.4–5.9)	5.6 (5.4–5.9)	5.6 (5.4–5.8)
Uric acid			
Uric acid ^†††^ (mg/dL)	5.2 (4.4–6.2)	6.3 (5.5–7.1)	4.8 (4.1–5.5) ***
Renal function			
Creatinine (mg/dL)	0.68 (0.57–0.83)	0.88 (0.79–0.98)	0.60 (0.54–0.67) ***
eGFR (mL/min/1.73 m^2^)	100.7 (87.5–116.4)	92.6 (81.3–105.0)	106.2 (92.4–122.2) ***
Albuminuria (mg/L)	8.7 (5.4–15.2)	8.4 (5.3–15.0)	9.0 (5.5–15.4) ***
Liver function			
AST (U/L)	23.0 (20.0–27.0)	24.0 (21.0–29.0)	22.0 (19.0–26.0) ***
ALT (U/L)	19.0 (14.0–27.0)	23.0 (17.0–33.0)	17.0 (13.0–23.0) ***
gGT (U/L)	17.0 (12.0–26.0)	22.0 (16.0–34.0)	14.0 (11.0–21.0) ***
Serum albumin (g/dL)	4.5 (4.4–4.7)	4.6 (4.4–4.7)	4.5 (4.3–4.6) ***
Total bilirubin (mg/dL)	0.6 (0.5–0.8)	0.7 (0.6–0.9)	0.6 (0.5–0.7) ***
Hematological parameters			
Leukocyte count (10^3^/μL)	5.7 (4.7–6.8)	5.9 (4.9–7.0)	5.6 (4.6–6.6)
Hematocrit (%)	41.6 (39–44.5)	44.9 (42.9–47.1)	40.0 (37.9–42.1) ***
Platelet count (10^3^/μL)	237.0 (202.0–276.0)	221.0 (190.0–256.0)	246.0 (211.0–286.0) ***
Red blood cell count (10^6^/μL)	4.7 (4.4–5.0)	5.1 (4.8–5.3)	4.5 (4.3–4.8) ***
Hemoglobin (g/dL)	13.7 (12.8–14.8)	15.1 (14.4–15.8)	13.1 (12.4–13.8) ***
Atherosclerotic risk factors			
Diabetes mellitus (%)	9.5%	12.4%	7.9% ***
Hypertension (%)	22.4%	30.9%	17.5% ***
Current smoking (%)	9.1%	20.3%	2.8%
Gout (%)	3.9%	9.8%	0.6% ***
Microalbuminuria (%)	11.3%	12.3%	10.8% *
Metabolic syndrome (%)	22.2%	26.9%	19.6% ***

Participants were analyzed after the exclusion of those with a history of ^†^ hypertension, ^††^ diabetes mellitus, ^†††^ gout, and # hyperlipidemia. Data are presented as medians (interquartile ranges). Abbreviations: BP, blood pressure; HDL, high-density lipoprotein; LDL, low-density lipoprotein; HbA1c, hemoglobin A1C; eGFR, estimated glomerular filtration rate; AST, aspartate aminotransferase; ALT, alanine aminotransferase; γ-GT, γ-glutamyl transferase. * *p* < 0.01, *** *p* < 0.0001 comparing female participants to male participants.

**Table 2 genes-13-01366-t002:** Association between *APOE*, *APOC1*, and *CLPTM1* gene region variants and serum lipid levels.

Genetic Variants	MM	Mm	Mm	*p* Value *	*p* Value **
*APOE* rs429358	TT (62,865)	CT (11,841)	CC (519)		
Total cholesterol # (mg/dL)	192.0 (170.0–215.0)	198.0 (176.0–222.0)	200.0 (181.0–224.0)	3.50 × 10^−66^	2.38 × 10^−71^
HDL cholesterol # (mg/dL)	53.0 (45.0–63.0)	52.0 (44.0–62.0)	51.0 (43.0–61.0)	7.83 × 10^−27^	1.21 × 10^−43^
LDL cholesterol # (mg/dL)	118.0 (98.0–139.0)	125.0 (105.0–146.0)	127.0 (108.0–149.0)	1.61 × 10^−110^	5.81 × 10^−120^
Triglyceride # (mg/dL)	89.0 (63.0–131.0)	94.0 (66.0–141.0)	100.0 (68.0–150.0)	8.00 × 10^−36^	4.27 × 10^−52^
*APOE* rs7412	CC (64,455)	TC (10,469)	TT (393)		
Total cholesterol # (mg/dL)	195.0 (173.0–218.0)	181.0 (161.0–204.0)	164.0 (137.0–199.0)	<10^−307^	<10^−307^
HDL cholesterol # (mg/dL)	53.0 (45.0–62.0)	55.0 (46.0–65.0)	54.0 (46.0–65.0)	2.34 × 10^−29^	2.37 × 10^−44^
LDL cholesterol # (mg/dL)	122.0 (103.0–143.0)	101.0 (85.0–120.0)	59.0 (46.0–74.0)	<10^−307^	<10^−307^
Triglyceride # (mg/dL)	89.0 (63.0–131.0)	94.0 (66.0–139.0)	114.0 (78.0–177.5)	2.42 × 10^−39^	1.45 × 10^−35^
*APOE- APOC1* rs439401	TT (26,681)	CT (36,300)	CC (12,458)		
Total cholesterol # (mg/dL)	194.0 (172.0–217.0)	192.0 (170.0–216.0)	192.0 (170.0–216.0)	1.11 × 10^−8^	1.16 × 10^−8^
HDL cholesterol # (mg/dL)	53.0 (45.0–63.0)	53.0 (45.0–63.0)	53.0 (45.0–63.0)	0.9970	0.3971
LDL cholesterol # (mg/dL)	121.0 (102.0–142.0)	118.0 (99.0–140.0)	116.0 (95.0–139.0)	1.28 × 10^−90^	2.20 × 10^−97^
Triglyceride # (mg/dL)	87.0 (62.0–127.0)	91.0 (64.0–133.0)	96.0 (67.0–142.0)	5.80 × 10^−57^	2.59 × 10^−66^
*APOC1* rs438811	CC (51,077)	TC (21,804)	TT (2258)		
Total cholesterol # (mg/dL)	194.0 (172.0–217.0)	190.0 (168.0–215.0)	188.0 (163.0–213.0)	1.68 × 10^−51^	1.63 × 10^−56^
HDL cholesterol # (mg/dL)	53.0 (45.0–63.0)	53.0 (45.0–63.0)	53.0 (45.0–63.0)	0.7682	0.6896
LDL cholesterol # (mg/dL)	121.0 (102.0–142.0)	115.0 (94.0–137.0)	106.0 (81.0–130.0)	<10^−307^	<10^−307^
Triglyceride # (mg/dL)	88.0 (63.0–129.0)	94.0 (65.0–139.0)	99.0 (68.0–149.0)	1.21 × 10^−67^	1.61 × 10^−85^
*CLPTM1* rs11672748	AA (20,021)	GA (37,145)	GG (17,842)		
Total cholesterol # (mg/dL)	193.0 (171.0–217.0)	193.0 (171.0–217.0)	192.0 (170.0–216.0)	0.0501	0.0223
HDL cholesterol # (mg/dL)	53.0 (45.0–62.0)	53.0 (45.0–63.0)	54.0 (45.0–63.0)	2.64 × 10^−13^	6.89 × 10^−16^
LDL cholesterol # (mg/dL)	120.0 (100.0–141.0)	119.0 (99.0–141.0)	118.0 (98.0–138.0)	2.06 × 10^−13^	2.83 × 10^−14^
Triglyceride # (mg/dL)	90.0 (64.0–132.0)	90.0 (63.0–132.0)	90.0 (64.0–132.0)	0.9639	0.9714
*CLPTM1* rs3786505	AA (20,031)	GA (37,183)	GG (17,840)		
Total cholesterol # (mg/dL)	193.0 (171.0–217.0)	193.0 (171.0–217.0)	192.0 (170.0–216.0)	0.0476	0.0187
HDL cholesterol # (mg/dL)	53.0 (45.0–62.0)	53.0 (45.0–63.0)	54.0 (45.0–63.0)	2.26 × 10^−13^	5.99 × 10^−16^
LDL cholesterol # (mg/dL)	120.0 (100.0–141.0)	119.0 (99.0–141.0)	118.0 (98.0–138.0)	1.33 × 10^−13^	1.67 × 10^−14^
Triglyceride # (mg/dL)	90.0 (64.0–132.0)	90.0 (63.0–132.0)	90.0 (64.0–132.0)	0.9535	0.9832

Data are presented as medians (interquartile ranges). Abbreviations: HDL, high-density lipoprotein; LDL, low-density lipoprotein. Number of the participants is presented in brackets after the genotypes. MM: homozygosity of major allele; Mm: heterozygosity; mm: homozygosity of minor allele. # Participants were analyzed after the exclusion of those with a history of hyperlipidemia. * *p* value: unadjusted; ** *p* value: adjusted for age, sex, BMI, and current smoking status.

**Table 3 genes-13-01366-t003:** Stepwise linear regression analysis including *APOE* region polymorphisms.

Lipid Profile	Total Cholesterol # (mg/dL)			LDL Cholesterol # (mg/dL)			HDL Cholesterol # (mg/dL)			Triglyceride # (mg/dL)		
	β	r^2^	*p* Value	β	r^2^	*p* Value	β	r^2^	*p* Value	β	r^2^	*p* Value
Sex (male vs. female)	0.0155	0.0050	1.25 × 10^−140^	--	--	--	0.0001	0.0001	0.0001	−0.0562	0.0191	1.10 × 10^−230^
Age (years)	0.0014	0.0361	<10^−307^	0.0015	0.0187	<10^−307^	0.0623	0.0873	<10^−307^	0.0031	0.0177	<10^−307^
Body mass index (kg/m^2^)	0.0018	0.0067	3.32 × 10^−119^	0.0051	0.0275	<10^−307^	−0.0094	0.1584	<10^−307^	0.0222	0.1475	<10^−307^
Current smoking (%)	0.0055	0.0004	3.18 × 10^−8^	--	--	--	−0.0242	0.0041	4.53 × 10^−88^	0.0818	0.0089	1.43 × 10^−182^
*APOE* rs7412 (CC vs. CT vs. TT)	−0.0298	0.0213	<10^−307^	−0.0897	0.0838	<10^−307^	0.0115	0.0019	2.07 × 10^−37^	--	--	--
*APOE* rs429358 (TT vs.TC vs. CC)	0.0101	0.0026	1.15 × 10^−47^	0.0172	0.0034	6.35 × 10^−65^	−0.0107	0.0016	1.30 × 10^−36^	--	--	--
*CLPTM1* rs3786505 (AA vs. AG vs. GG)	--	--	--	−0.0044	0.0007	3.10 × 10^−15^	--	--	--	--	--	--
*CLPTM1* rs11672748 (AA vs. AG vs. GG)	--	--	--	--	--	--	0.0037	0.0006	1.48 × 10^−15^	--	--	--
*APOC1* rs438811 (CC vs. CT vs. TT)	--	--	--	--	--	--	--	--	--	0.0210	0.0041	7.54 × 10^−34^
*APOE- APOC1* rs439401 (TT vs.TC vs. CC)	--	--	--	--	--	--	--	--	--	0.0101	0.0006	4.78 × 10^−14^

# Participants were analyzed after the exclusion of those with a history of hyperlipidemia.

**Table 4 genes-13-01366-t004:** Logistic regression analysis for metabolic syndrome, including *APOE* and *APOC1* genotypes.

Metabolic Syndrome	β	SE	*p* Value	OR	95% CI
Sex (male vs. female)	0.0891	0.0210	2.12 × 10^−5^	1.09	1.04–1.14
Age (years)	0.0601	0.0010	<10^−307^	1.06	1.06–1.06
Body mass index (kg/m^2^)	0.3315	0.0031	<10^−307^	1.39	1.38–1.40
Current smoking (%)	0.5271	0.0329	7.58 × 10^−58^	1.69	1.59–1.81
*APOE* rs429358 (TT vs.TC vs. CC)	0.1830	0.0240	2.29 × 10^−14^	1.20	1.15–1.26
*APOC1* rs438811 (CC vs. CT vs. TT)	--	--	--	--	--

OR, odds ratio; 95% CI, 95% confidence interval; SE, standard error; *p* value: adjusted for age, sex, body mass index, and current smoking status.

## Data Availability

The data presented in this study are available on request from the corresponding author.

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
