# Peer review of "Genetic Variants at the APOE Locus Predict Cardiometabolic Traits and Metabolic Syndrome: A Taiwan Biobank Study"

_genes, 2022, doi:10.3390/genes13081366_

Round 1
Reviewer 1 Report
Yeh et al. prepared an interesting and valuable article related to such genetic aspects of the pathogenesis of dyslipidemia. The subject is very important because obesity, metabolic syndrome, and its complications are the most important medical problems worldwide. The paper is generally well prepared. However, I think some changes are needed, which may further improve the quality of the manuscript.
Although the introduction gives sufficient information on the molecular aspects and properties of apoE, there is information on metabolic syndrome and cardiovascular disease. It should be shortly explained, why lipid metabolism disturbances are so important. Dyslipidemia is associated with elevated lipid parameters and/or the presence of modified lipoproteins. Dyslipidemia is a modifiable risk factor of atherosclerosis, which is currently the most important pathological mechanism leading to the development of cardiovascular disease. The most important forms of atherosclerotic cardiovascular disease are coronary heart disease, cerebrovascular disease, and peripheral arterial disease. (You can see for example the following high-quality and up-to-date literature positions: doi.org/10.3390/antiox11050856; doi.org/10.3390/ijerph17249339; doi.org/10.3390/nu13113843; doi.org/10.3390/nu12092671; doi.org/10.1155/2015/912047). (The literature is not compulsory to be cited)
It should be “HbA1c”, not “HbA1C”.
All continuous variables were presented as median and interquartile range or only ones without normal distribution? For variables with normal distribution mean and standard deviation would be more appropriate. (lines 158-160)
“liver function parameters of AST, ALT, γ-GT, albumin, and total bilirubin levels” (line 180): In my opinion, it should be “liver injury and function parameters”. Elevated AST and ALT levels are associated with liver injury, but say nothing about liver function. Such parameters are useful in the assessment of liver function: albumin, bilirubin, and INR.
Reviewer 2 Report
The authors submitted a research article in which they explored the genetic determinants of cardiometabolic traits and metabolic syndrome at the APOE locus in a Taiwanese population. They enrolled 81,387 participants from the Taiwan Biobank and investigated the genetic determinants
of cardiometabolic traits and metabolic syndrome by using regional association analysis. The authors found that APOE ɛ2, ɛ3 and ɛ4 variants, defined by the APOE rs7412 and APOE rs429358
genotypes, were the strongest genetic determinants in the APOE locus of total, LDL, and HDL cholesterol levels in Taiwan. The authors suggested that CLPTM1 as a novel candidate locus for LDL
and HDL cholesterol levels independent of the APOE variants in an Asian population. The aim of the study is clear and deserves to be investigated. The manuscript has logical structure and composes of well-balansed sub-sections covering all aspects of the study and findings. The tables and figure are legible and clear. The conclusive part seems to be informative. The strengths of the study is a large population, the weaknesses are a lack of GWAS evaluation and retros[ective design. Overall, the article is well written and add sufficient knowledge to previous one. I congratulate the authors on the study. However, I have some issues to discuss.
1. Table 1. Please, add a column with data for entire population of the patients.
2. Please, give a brief explanation od ethnic groups that were included in the study.
3. More concise recommendation about clinical significance requires.
Round 2
Reviewer 1 Report
The paper has been improved. I recommend it for publication in its current form.